# Arterial Thrombosis in an Unusual Site (Ulnar Artery) after COVID-19 Vaccination—A Case Report

**Aurelio Sessa** [1],*, **Marco Gattamorta** [2], **Maurizia Punginelli** [3] **and Gianluigi Maggioni** [3]

1 Internal Medicine Unit, Isber Clinic, I-21100 Varese, Italy
2 Radiolgy Unit, Isber Clinic, I-21100 Varese, Italy; info@medicinaisber.it
3 Pharmaceutical Service, Territorial Health Service of Insubria, I-21100 Varese, Italy; punginellim@ats-insubria.it (M.P.); servizio.farmaceutico@ats-insubria.it (G.M.)
* Correspondence: aurelio.sessa@libero.it; Tel.: +39-0332-242971

**Abstract:** Spontaneous events have been reported after COVID-19 vaccination. In this case, we report a thrombotic event in an unusual site (ulnar artery) after COVID-19 vaccination. The patient (69 year-old-male) had no changes after a laboratory investigation regarding thrombophilic pattern, but nevertheless had atherothrombotic predisposing conditions. Arterial thrombotic events have more frequently been reported after mRNA vaccines than after adenovirus vaccines. This is the first case reported of thrombosis of the ulnar artery occurring in the same side of the body where the vaccination took place. However, it must be noted that COVID-19 vaccines cumulatively offer a net positive effect, despite rare adverse effects.

**Keywords:** COVID-19 vaccination; thrombotic events; arterial thrombosis

## 1. Background

COVID-19 vaccination has reduced the diffusion of infection and related complications around the world [1]. However, spontaneous reports of thrombotic events have been reported after COVID-19 vaccination. The first cases were after the Oxford AstraZeneca vaccine (ChAdOx1-S AZD1222), which was then temporarily suspended by some European countries. However, thrombotic events, both venous and arterial, have been reported after other vaccines, including mRNA (Pfizer/BioNTech and Moderna/Biotech) [2]. Nevertheless, the benefits of the vaccine in the fight against COVID-19 have been universally recognized.

## 2. Case Presentation

Four days after having received the third dose (0.25 mL) of the Spikevax vaccine (mRNA-1273, Moderna/Biotech, Spain), after a previous two doses of BNT 162b2 (Pfizer–BioNTech, Belgium and Germany), a 69-year-old man presented with a 4 cm redness swelling on the hypothenar eminence of the left hand, the same side of the site (left) of vaccination. The swelling was painless, with only a small amount of pain under pressure, with a consistency like rope (Figure 1).

The patient, a radiology Technician, showed the hand to the Radiologist that worked with him in the Radiology Unit and a musculoskeletal ultrasound was performed, using greyscale and Doppler imaging. Consequently, a thrombosis of the arcuate branch of the ulnar artery measuring 30 mm was identified (Figure 2).

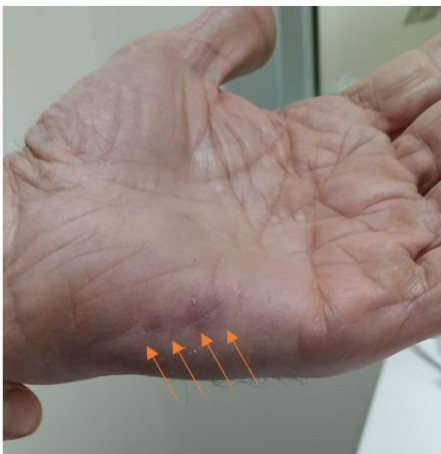

**Figure 1.** Redness swelling in hypothenar eminence of left hand (the arrows indicate the lesion).

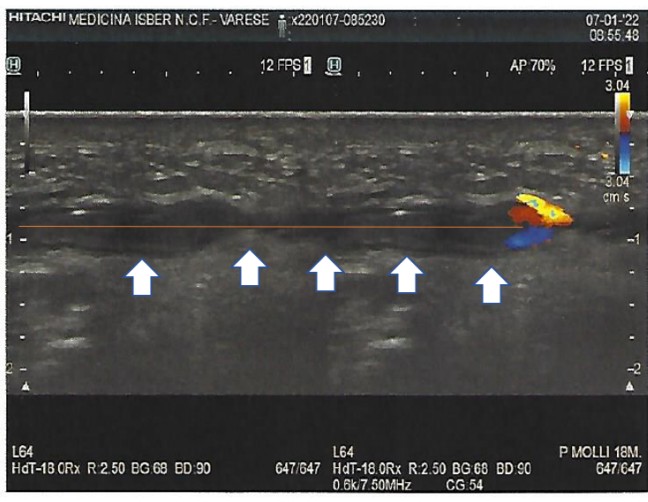

(**a**)

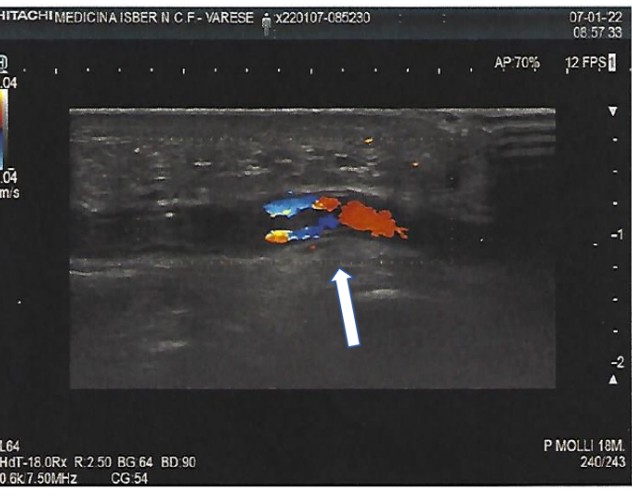

(**b**)

**Figure 2.** Subocclusive arterial thrombosis of the arcuate branch of the ulnar artery. The arrows indicate the thrombosis in all its length (**a**); Doppler image shows how the blood flow spreads by crossing the intraluminal thrombus (arrow) (**b**).

No particular events, such trauma or repeated pressure actions, were reported in the previous days. The patient was then evaluated by the internist, who took note of the clinical picture. On physical examination, the patient had a blood pressure rate of 133/82 mmHg, a heart rate of 70, and oximetry of 99%. His BMI was 23.4 kg per m$^2$. He had no signs of distal ischemia, and had a normal arterial pulse. The patient had a history of hypothyroidism after total thyroidectomy for goiter and benign prostatic hyperplasia, and undertook daily therapy with l-thyroxine 150 mcg and tamsulosin 0.4 mg. The patient's last blood investigation was September 2021, with blood count, glucose, creatinine, alanine transaminase (ALT), aspartate transaminase (AST), prostate-specific antigen (PSA) and thyroid-stimulating hormone (TSH) rates all in the range of normality. The lipid panel showed low-density lipoprotein cholesterol (LDL-C) of 109.8 mg/dL (total cholesterol 188 mg/dL, HDL-cholesterol 50 mg/dL and triglycerides 141 mg/dL). He never smoked cigarettes.

The patient did not report taking any products such as herbs or homeopathic substances. In the absence of clinical conditions that could justify this arterial thrombosis, this was identified as an adverse drug reaction (ADR) to the vaccine.

## 3. Investigations

Doppler imaging of the arm and hand highlighted an extensive thrombus in the arcuate branch of the ulnar artery (Figure 2), showing that the blood flow was spread across the intraluminal thrombus.

The results of the blood investigations ruled out the secondary cause of thrombosis or immune disorders (Table 1).

**Table 1.** Laboratory findings.

| Test | Value | Normal Range |
|---|---|---|
| Hemoglobin (g/L) | 143 | 130–175 |
| White blood cells (per mm$^3$) | 6200 | 4800–10,800 |
| Platelet count (per mm$^3$) | 244,000 | 130,000–400,000 |
| Prothrombin time (PT) (INR) | 1.0 | 0.8–1.2 |
| Activate partial thromboplastine time (aPTT) (sec) | 32.2 | 22.7–32.5 |
| D-dimers (mcg/mL) | 0.25 | <0.50 |
| Lupus anticoagulant | 0.50 | <1.20 |
| Antiphospholipid antibodies IgG (U/mL) | 3.1 | <12 |
| SARS-CoV-2 RT-PCR test | Negative | |

## 4. Differential Diagnosis

The patient did not report any traumatic event. Therefore, this excluded one of the most frequent causes of ulnar thrombosis (hypothenar hammer syndrome): thrombosis developing after repetitive trauma to the hypothenar eminence [3].

Although arterosclerosis is a common condition, a first thrombotic event at this site is unusual [4]. Moreover, the patient had no cardiovascular risk factors.

## 5. Treatment

The internist suggested taking 100 mg acetylsalicilic acid once a day for at least 30 days.

## 6. Outcome and Follow-Up

The event was reported to the Italian Agency of Drugs (AIFA) as a suspected ADR [5] (report number V-202202-03F459-8).

Ten days afterwards, a Doppler ultrasound showed a reduction in the dimension of the thrombosis (22.9 mm vs. 30.0 mm) and a recanalization of the thrombus (Figure 3). The patient remained under the care of Internal Medicine consultant. The probable link between the vaccine and thrombosis of the ulnar artery was hypothesized and discussed with the territorial senior Pharmacist.

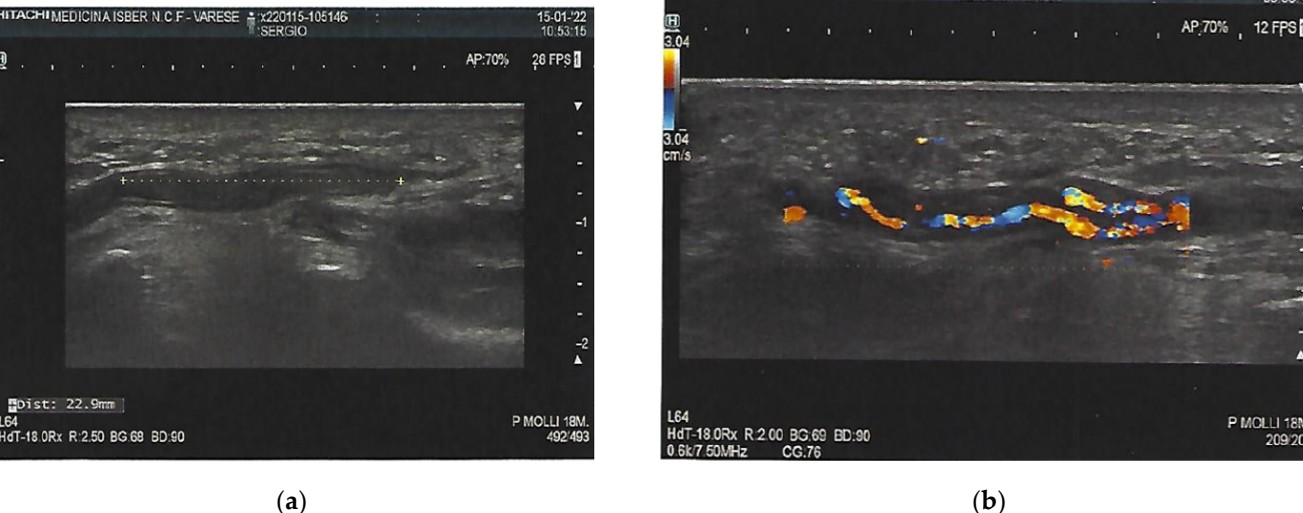

(**a**) (**b**)

**Figure 3.** Reduction in the length of the thrombus after 10 days (**a**); Doppler ultrasound after 10 days with reduction in the length and thickness of the thrombus (22.9 mm vs. 30.0 mm) (**b**).

## 7. Discussion

To fight the COVID-19 pandemic, the European Regulatory Agency approved five vaccines by December 2021: BNT 162b2 (Pfizer–BioNTech, Belgium), AD26 (Janssen JNJ, Netherlands), mRNA-1273 (Moderna NIAID, Spain), ChAdOx1 (AstraZeneca, Oxford University, UK) and NXV-CoV2373 (Novavax, MD, USA). Thrombotic complications rarely occur in the general population underwent vaccination against SARS-CoV-2. Few cases have been described after the ChAdOx1 nCoV-19 [6] and AD26.COV2.S vaccines [7]. These reports ranged from severe thrombocytopenia, bleeding, and arterial thrombosis to venous thrombosis in unusual anatomical sites (cerebral venous sinus thrombosis, or thrombosis in the portal, splanchnic, or hepatic veins) occurring within 5 to 24 days of vaccination [8]. Of these, ChAdox1 nCov-19 (AstraZeneca) was associated with vaccine-induced thrombosis and thrombocytopenia (VITT). VITT typically manifests with venous thrombosis (often cerebral), and a few arterial thromboses were also noted [9]. Thrombotic complications are variously associated with changes in coagulation tests (aPTT, PT, low platelet count) [10]. Young women aged ≤50 years are the most exposed to thrombotic events, and no risk factors for thrombosis or thrombophilia have been reported for this demographic; the European Medicine Agency (EMA) evaluated the incidence of thrombosis as being 1 in 125,000 [11]. The mechanism responsible for VITT was similar to that for heparin-induced thrombocytopenia (HIT) [12]. Thus, the term VITT has been proposed to describe this condition. In this case report, the relationship between event and vaccine was defined as probable according to the Naranjo algorithm (score 7) [13]. In an anti-SARS-CoV-2 vaccination descriptive analysis of thrombotic risk reported to the World Health Organization (WHO) Global Database for Individual Case Safety Reports (VigiBase), spontaneous reports of thrombotic events were found in 1197 persons for Comirnaty, 325 for Moderna, and 639 for AZD1222 [2]. They evaluated the reporting rate for venous and arterial thrombotic event cases during the time period after vaccination. For arterial thrombotic events, the rate was 0.13 (95% CI 0.12–0.14) cases per million vaccinated person-days. It is interesting to highlight that an imbalance was recorded between arterial thrombotic events in mRNA vaccines (67.9% (813/1197) for Comirnaty and 77.6% (253/325) for Moderna). Conversely, for AZD1222, the proportion of venous thrombotic events were more frequent than arterial thrombotic events [(52.2% (334/639) vs. 48.2% (308/639)]. The time frame between vaccination and the arterial thrombotic event was the same for the three vaccines (Comirnaty, Moderna and AZD1222) (median of 2 days), although a significant difference was identified between AZD1222 (median 6 days) and both mRNA vaccines (median of 4 days). Searching the spontaneous reports (COVID-19 MRNA VACCINE MODERNA (CX-024414))

collected in EudraVigilance, the EU database used for monitoring and analysing suspected side effects [14], as of 26 February 2022, a total of 243,789 cases of suspected side effects with Spikevax were spontaneously reported from EE/EEA countries. Searching "*arterial thrombosis*", there are 16 cases reported, 9 in the 18–64 years age group (6F and 3M), and 7 in those aged 68–85 years (3F and 4M). The outcome of these 15 case were: 1 fatal; 6 not resolved; 2 resolved; 2 resolved with sequelae; 2 resolving; and 3 unknown.

## 8. Site of Thrombosis

In a systematic review, the thrombosis location was found to have the potential to affect more vessels in each patient [15], although two studies reported patients with only one site of thrombosis. Most thrombosis sites were venous, and the arterial sites were aortic vessels, aorto-iliac, or the internal carotid artery.

## 9. Learning Points/Take-Home Messages

It is possible that arterial thrombotic events after COVID-19 vaccination can occur in unusual sites other than those that are well known. This is of importance as clinicians must be vigilant for cardiovascular complications, not just in the acute or convalescent phases of COVID-19 infection, but also after a seemingly innocuous vaccination against the virus [16].

## 10. Patient's Perspective and General Considerations

The patient, although reassured of the benign evolution, is worried about a possible fourth dose of the COVID-19 vaccine in light of this. A possible thrombotic event at a vital site (heart or brain) could cause significant damage. The documentation produced on what has happened can be presented to evaluate a possible exemption.

In these situations, a lack of guidance exists on how to manage a patient experiencing such an effect of a vaccination. It is recommended that patients do not take a further dose if they have experienced these effects [17].

It is also important to highlight that COVID-19 vaccines still cumulatively offer a net positive effect to the individual and to the population, and that rare adverse effects should not be a justification for avoiding vaccination.

## 11. Comment

This is the first case reported of a thrombosis of the ulnar artery. Luckily, the arterial thrombosis was not in a vital organ, and could be resolved. Moreover, curiously, it occurred on the same side (left) of the body as the vaccination took place.

**Author Contributions:** Conceptualization, A.S.; methodology, A.S., M.G., M.P. and G.M.; formal analysis and validation, A.S., M.G., M.P. and G.M.; investigation, A.S. and M.G.; writing, A.S.; writing—review and editing, A.S., M.G., M.P. and G.M. All authors have read and agreed to the published version of the manuscript.

**Funding:** This research received no external funding.

**Institutional Review Board Statement:** Not applicable.

**Informed Consent Statement:** Informed consent was obtained from the patient involved in this case report.

**Conflicts of Interest:** The authors declare no conflict of interest.

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
