# Peer review of "Arterial Thrombosis in an Unusual Site (Ulnar Artery) after COVID-19 Vaccination—A Case Report"

_clinpract, doi:10.3390/clinpract12030028_

Round 1

Reviewer 1 Report

Content suggestions:

  1. I would like to kindlyask the authors whether they know if the patient was smoker, and how was his blood pressure ?
  2. Do they know the levels of the markers in lipidogram ?
  3. What are the recommendations of the authors for the prevention of such events in the future ?

The article can be published after incorporation of the response to the comments.

Author Response

dear reviewer,

thank you about your questions

The patients was not a smoker (he never smoked). His blood pressure was normal.

During the visit by internist he had 133/82 mmHg and pulse rate 70.

About his lipid profile,  the patient had blood examination in september 2021 with cholesterol 188 mg/dl, HDL-cholesterol 50 mg/dl, triglycerides 141 mg/dl, LDL-cholesterol 109,8  

This case suggests the recommendation to follow up all patients after COVID-19 vaccination presenting any adverse reactions.

We'll add your suggestions in the new version of the manuscript

thanks! 

Reviewer 2 Report

In this manuscript, the authors reported one arterial thrombosis case after the second dose of Spikevax vaccine. The report is interesting; however, all kind of lab results indicate that the platelet activity or coagulation is not elevated after vaccination. I have one major concern.

Is the previous blood test lab result of this patient available? This evidence may make the reader better understand the case.

Author Response

dear reviewer

thank you for your questions

The patients received the booster (third) dose of Spikevax after two doses of Pfizer.

He made in september 2021 some general blood tests with RBC, WBC and PLTs in the range of normality as well as glucose, creatinine, AST, ALT, total cholesterol, HDL-cholesterol, triglycerides, LDL-cholesterol, PSA and TSH.

He didn't have any other blood test (e.g. coagulation) because he had no health problems other than hypothyroidism and benign prostatic hyperplasia, reported in the description of the case

We'll add in the new version of the manuscript your suggestions

Thanks